# Organophotocatalytic selective deuterodehalogenation of aryl or alkyl chlorides

Yanjun Li [1], Ziqi Ye[1], Yu-Mei Lin [1], Yan Liu[1], Yumeng Zhang[1] & Lei Gong [1]✉

Development of practical deuteration reactions is highly valuable for organic synthesis, analytic chemistry and pharmaceutic chemistry. Deuterodehalogenation of organic chlorides tends to be an attractive strategy but remains a challenging task. We here develop a photocatalytic system consisting of an aryl-amine photocatalyst and a disulfide co-catalyst in the presence of sodium formate as an electron and hydrogen donor. Accordingly, many aryl chlorides, alkyl chlorides, and other halides are converted to deuterated products at room temperature in air (>90 examples, up to 99% D-incorporation). The mechanistic studies reveal that the aryl amine serves as reducing photoredox catalyst to initiate cleavage of the C-Cl bond, at the same time as energy transfer catalyst to induce homolysis of the disulfide for consequent deuterium transfer process. This economic and environmentally-friendly method can be used for site-selective D-labeling of a number of bioactive molecules and direct H/D exchange of some drug molecules.

[1] Key Laboratory of Chemical Biology of Fujian Province, iChEM, College of Chemistry and Chemical Engineering, Xiamen University, Xiamen 361005 Fujian, China. ✉email: gongl@xmu.edu.cn

euterium-containing molecules are widely used in investigation of reaction mechanisms[1–7], organic synthesis[8–12], analytic chemistry[13–15], and pharmaceutical chemistry[16–19]. In particular, D-labeling techniques have been established as an important tool in drug development because incorporation of deuterium atoms into bioactive compounds can dramatically affect their metabolic and pharmacokinetic properties[20,21]. In 2017, Austedo (deutetrabenazine) was approved by FDA as the first deuterated drug used for the treatment of chorea, a dyskinesia associated with Huntington's disease[22]. Our knowledge of the distribution, metabolism and excretion properties of a number of drugs such as Tolperisone, Rofecoxib and Dasatinib has been remarkably enhanced by deuteration (Fig. 1a)[23,24]. Development of practical and bio-friendly deuteration protocols, in particular those being applicable to structurally complex pharmaceutical compounds, is of great interests and in high demand[25–37].

Deuterodehalogenation of organohalides is a straightforward strategy with which to incorporate deuterium atoms at specific positions. However, the available methods often rely on transition-metal catalysis and suffer from harsh reaction conditions, narrow substrate scope, and the use of toxic and expensive deuterium sources[38]. Achieving a high level of deuterium incorporation is also a challenging task[39]. These factors apparently limit their application in pharmaceutic and life science. Recently, several subtle strategies have been developed to avoid use of transition metal catalysts[40–44]. For example, Renaud et al. reported thiol-catalyzed deuterative deiodination of alkyl iodides with $D_2O$ as a source of deuterium atoms with a stoichiometric amount of $Et_3B$ as the reductant[40]. Liu et al. developed potassium methoxide/disilane-mediated deuterodehalogenation of (hetero) aryl bromides or iodides in $CD_3CN$[41]. Loh et al. developed the photocatalytic deuterodehalogenation of aryl halides using porous CdSe nanosheets as the catalyst and $D_2O$ as the deuteration reagent[42]. Organic chlorides tend to be less expensive, more abundant and readily available than other halides. There are however very limited methods for deuterodehalogenation of aryl chlorides[45,46], and to date, deuterodehalogenation of unactivated alkyl chlorides has not been reported. Several significant challenges are associated with the strategy, including (i) the highly negative reduction potentials of unactivated C-Cl bonds (for example, $E_{red}$ [$(CH_3CH_2CH_2Cl)/(CH_3CH_2CH_2Cl)^{·-}$] = −2.8 V vs SCE)[47–52], (ii) the strong possibility of side reactions such as elimination in transition metal catalysis[53], (iii) the lack of an effective approach to deuterium transfer which can ensure a high incorporation of deuterium[29]. Moreover, precise differentiation of C–Cl bonds in polychlorinated compounds that have little or no difference in polarity or other features is highly useful in multi-step organic syntheses of deuterated biomolecules, but remains an unsolved problem (Fig. 1b).

Our group has long had an interest in the design of economic and environmentally-friendly photochemical synthesis[54–58]. We questioned whether implementing reducing organophotoredox catalysis could initiate the cleavage of unactivated aryl/alkyl C–Cl bonds, and well-matched organocatalysis could bridge the energy gap between the O–D bonds in $D_2O$ (BDE = 118 kcal mol$^{−1}$) and C–D bonds in deuterated hydrocarbons (BDE = 88-110 kcal mol$^{−1}$)[29]. This would allow us establish an effective channel for deuterium transfer, suppressing side reactions, and developing a general and convenient D-labeling method.

Herein, we report photocatalytic deuterodehalogenation for both aryl and alkyl chlorides by synergistic aryl-amine-based organophotoredox and disulfide organocatalysis in the presence of sodium formate as a mild electron and hydrogen sacrificial agent. This reaction provides a practical access to structurally diverse deuterated products including many bioactive molecules and drugs, and exhibits excellent site-selectivity in deuterodehalogenation of polychlorinated compounds which would be beneficial for the assembly of libraries of deuterated compounds (Fig. 1c).

## Results

**Reaction development.** Aryl amines have been established as useful photoredox catalysts in some light-induced reactions[59,60]. We questioned whether adjusting structures of conjugated aryl amines would enable us find effective visible-light photocatalysts for deuterodehalogenation of organic chlorides. For example, 1,3,5-tris(4′-(N,N-dimethylamino)phenyl)benzene (**PC1**, Fig. 2) has been intensively used as chemo-sensors[61]. To investigate their applications in synthetic chemistry, photophysical and electrochemical properties of **PC1** and **PC2–PC4** were measured and summarized in Fig. 2, including the maximum absorption, excitation, the energies of the first singlet excited state ($E_{0,0}$), and the oxidation potentials at excited state ($E^*_{ox}$). These compounds exhibited obvious visible-light absorption (see more details in Supplementary Fig. 15), long excited-state lifetimes (2–24 ns), and strongly reducing ability ($E^*_{ox} = −1.61$ to $−2.71$ V vs SCE), which allowed them serve as catalyst candidates being capable of cleaving unactivated C(sp$^2$)-Cl and C(sp$^3$)-Cl bonds ($E_{red} = −1.8$ to $−3.0$ V), and initiating radical process under visible light conditions.

We embarked on an examination of photocatalytic deuterodehalogenation reactions by using **PC1** (5 mol%) as a photocatalyst, propane-1-thiol (30 mol%) as the co-catalyst for deuterium atom transfer[25,62], 4-chloro-1,1′-biphenyl (**1a**, $E_{red} = −2.30$ V vs SCE in DMSO) as the model substrate, and $D_2O$ as the deuterium source. Upon irradiation with a 50 W blue LEDs lamp ($λ_{max} = 400$ nm) at room temperature, the reaction of **1a** with excess $D_2O$ in DMSO produced only a trace amount of **2a** (Fig. 3, entry 1). The reaction could be improved to some extent by the addition of inorganic bases, which provided low to moderate conversion and deuterium incorporation (entries 2–4). It was thought that introduction of certain reductive hydrogen donor could tune the reactivity of the catalytic cycles and improve the efficiency of the reaction. The Hantzsch ester was found to be inappropriate for this transformation (entry 5), but formate salts accelerated the reaction significantly and led to an obvious increase in the deuterium incorporation (entries 6–8). Sodium formate was identified as the best additive, delivering in 15 h the product (**2a**) in quantitative yield and with 85% deuterium incorporation (D-inc.) (entry 6). The result of using sodium formate to promote the transfer of electrons and hydrogen atoms is fully consistent with the observation in the intermolecular addition of aryl and heteroaryl radicals to enecarbamate substrates reported recently by Jui et al.[63]. Other sulfur-containing cocatalysts such as ethyl 2-mercaptoacetate (entry 9), 4-methoxybenzenethiol (entry 10), dicyclohexyl disulfide (entry 11), dipropyl disulfide (entry 12), dimethyl disulfide (entry 13), and diethyl disulfide were examined. The D-incorporation was significantly improved (92% D-inc.) by replacing the thiol with dimethyl disulfide (entry 13). The higher reaction efficiency might be attributed to the fact that in contrast to the thiols, organic disulfides can be easily cleaved into free thiyl radicals useful in the deuterium transfer without introduction of additional protons[64]. Photocatalysts **PC2–PC4** could be used in the reaction, while gave reduced yields or/and deuterium incorporation (entries 16–18). Control experiments revealed that the photocatalyst and the light irradiation were essential for the reaction to proceed (entries 19, 20), and the sulfur-based cocatalyst was critical for achieving a high level of deuterium incorporation (entry 21).

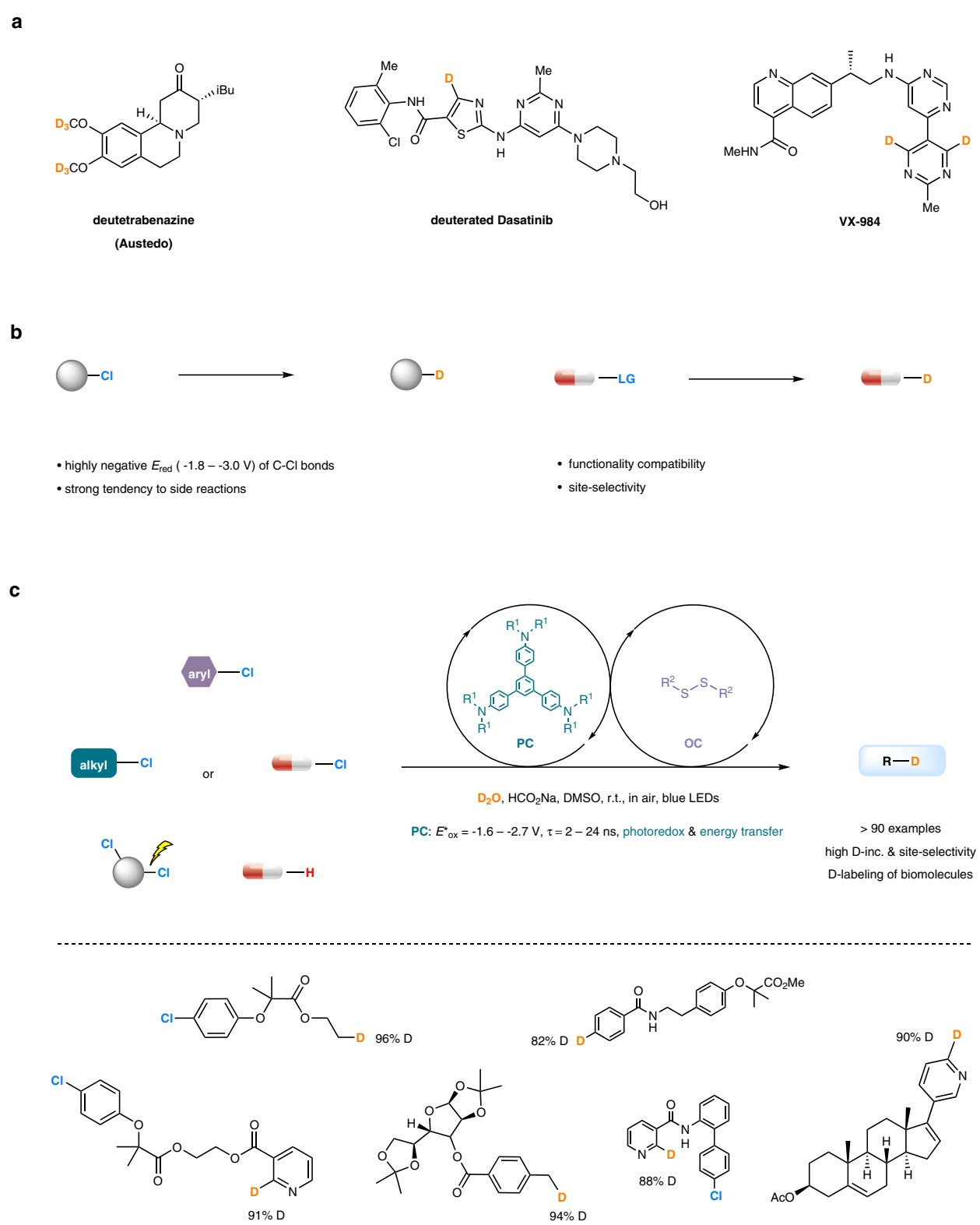

**Fig. 1 Overview of this work. a** Deuterated drug molecules. **b** Challenges in deuterodehalogenation of organic chlorides and its application to drug labeling. **c** This work: organophotocatalytic selective deuterodehalogenation of aryl or alkyl chlorides. LG, leaving group; PC, photocatalyst; OC, organocatalyst; r.t., room temperature; D-inc., deuterium incorporation; DMSO, dimethyl sulfoxide.

**Reaction scope of aryl chlorides and other halides**. With the optimal conditions in hand, the substrate scope of the photo-catalytic deuterodehalogenation of aryl chlorides was investigated (Fig. 4). Products bearing a phenyl substituent (**2a**), a fused ring

(**2b**, **2c**) or an electron-withdrawing group (**2d–2f**) were formed within 10–12 h from the chlorobenzene derivatives (67–86% yield, 83–97% D-inc.). 1-Chloro-4-methoxybenzene failed to provide the desired product (**2g**) when using **PC1** as the

| entry | photocatalyst | absorption $\lambda_{max}$ (nm) | excitation $\lambda_{max}$ (nm) | excited state lifetime (ns) | $E_{0,0}$ (eV) (from $\lambda_{max}$) | $E_{0,0}$ (eV) (from $\lambda_{onset}$) | $E^*_{ox}$ (V vs SCE) |
|---|---|---|---|---|---|---|---|
| 1 | **PC1** | 315 | 440 | 10 | 2.82 | 3.39 | -1.95 − -2.52 |
| 2 | **PC2** | 352 | 426 | 2 | 2.91 | 3.38 | -1.61 − -2.08 |
| 3 | **PC3** | 334 | 491 | 15 | 2.53 | 3.00 | -2.24 − -2.71 |
| 4 | **PC4** | 333 | 453 | 24 | 2.74 | 2.92 | -2.42 − -2.60 |

**Fig. 2 Photophysical and electrochemical properties of arylamine-based photocatalysts PC1 − PC4.** All the data are measured in DMSO (0.10 or 1.0 mM) solution at room temperature. PC, photocatalyst; DMSO, dimethyl sulfoxide.

| entry | PC | RSH or RSSR | additive | t (h) | conv. (%)[a] | D-inc. (%)[b] |
|---|---|---|---|---|---|---|
| 1 | **PC1** | $n$PrSH | none | 15 | < 5 | n.d. |
| 2 | **PC1** | $n$PrSH | $Na_2CO_3$ | 15 | 77 | 20 |
| 3 | **PC1** | $n$PrSH | NaOAc | 15 | 60 | 10 |
| 4 | **PC1** | $n$PrSH | $Na_3PO_4$ | 15 | 57 | 16 |
| 5 | **PC1** | $n$PrSH | Hantzsch ester | 15 | 0 | n.a. |
| 6 | **PC1** | $n$PrSH | $HCO_2Na$ | 15 | quant. | 85 |
| 7 | **PC1** | $n$PrSH | $HCO_2K$ | 15 | quant. | 84 |
| 8 | **PC1** | $n$PrSH | $HCO_2NH_4$ | 15 | quant. | 73 |
| 9 | **PC1** | ethyl 2-mercaptoacetate | $HCO_2Na$ | 9 | quant. | 53 |
| 10 | **PC1** | 4-MeOPhSH | $HCO_2Na$ | 9 | 90 | 24 |
| 11 | **PC1** | $(CyS)_2$ | $HCO_2Na$ | 9 | 69 | 6 |
| 12 | **PC1** | $(nPrS)_2$ | $HCO_2Na$ | 9 | quant. | 91 |
| 13 | **PC1** | $(MeS)_2$ | $HCO_2Na$ | 9 | quant. | 92 |
| 14 | **PC1** | $(EtS)_2$ | $HCO_2Na$ | 9 | 85 | 92 |
| 15 | **PC1** | $(nBuS)_2$ | $HCO_2Na$ | 9 | 66 | 9 |
| 16 | **PC2** | $(nPrS)_2$ | $HCO_2Na$ | 9 | 26 | 92 |
| 17 | **PC3** | $(nPrS)_2$ | $HCO_2Na$ | 9 | 36 | 84 |
| 18 | **PC4** | $(nPrS)_2$ | $HCO_2Na$ | 9 | 51 | 81 |
| 19 | none | $(MeS)_2$ | $HCO_2Na$ | 25 | < 5 | n.d. |
| 20[c] | **PC1** | $(MeS)_2$ | $HCO_2Na$ | 25 | 0 | n.d. |
| 21 | **PC1** | none | $HCO_2Na$ | 25 | 90 | 10 |

**Fig. 3 Initial experiments for photocatalytic deuterodehalogenation of aryl chlorides.** Reaction conditions: **1a** (0.10 mmol), $D_2O$ (0.10 mL), **PC1**−**PC4** (0.0050 mmol), thiol or disulfide (0.030 mmol), additive (0.20 mmol), DMSO (0.50 mL), a 50 W blue LEDs lamp ($\lambda_{max} = 400$ nm), room temperature (~ 30 °C under irradiation), in air. [a]Degree of conversion determined by $^1$H-NMR. [b]D-inc. determined by $^1$H-NMR. [c]In the dark. PC, photocatalyst; eq, equivalent; r.t., room temperature; conv., conversion; D-inc., deuterium incorporation; t, reaction time; quant., quantitative conversion; n.d., not determined; n.a., not applicable; DMSO, dimethyl sulfoxide.

photocatalyst, perhaps because the electron-rich chlorobenzene contains the C(sp²)-Cl bond with higher reduction potential ($E_{red} = −2.90$ V vs SCE in DMSO). This transformation could be improved by replacing **PC1** by **PC4** of stronger reducing ability ($E^*_{ox} = −2.42$ to $−2.60$ V), delivering **2g** in 74% yield and with 63% D-inc. Aromatic heterocycles are common in organic

compounds including natural products and bioactive molecules, and a number of pharmaceutically important heterocyclic chlorides were examined and found to be well tolerated under the standard conditions. For example, deuterated heterocycles including pyridines with an electron-donating substituent (product **2i**) or an electron-withdrawing substituent (products **2j**–**2o**),

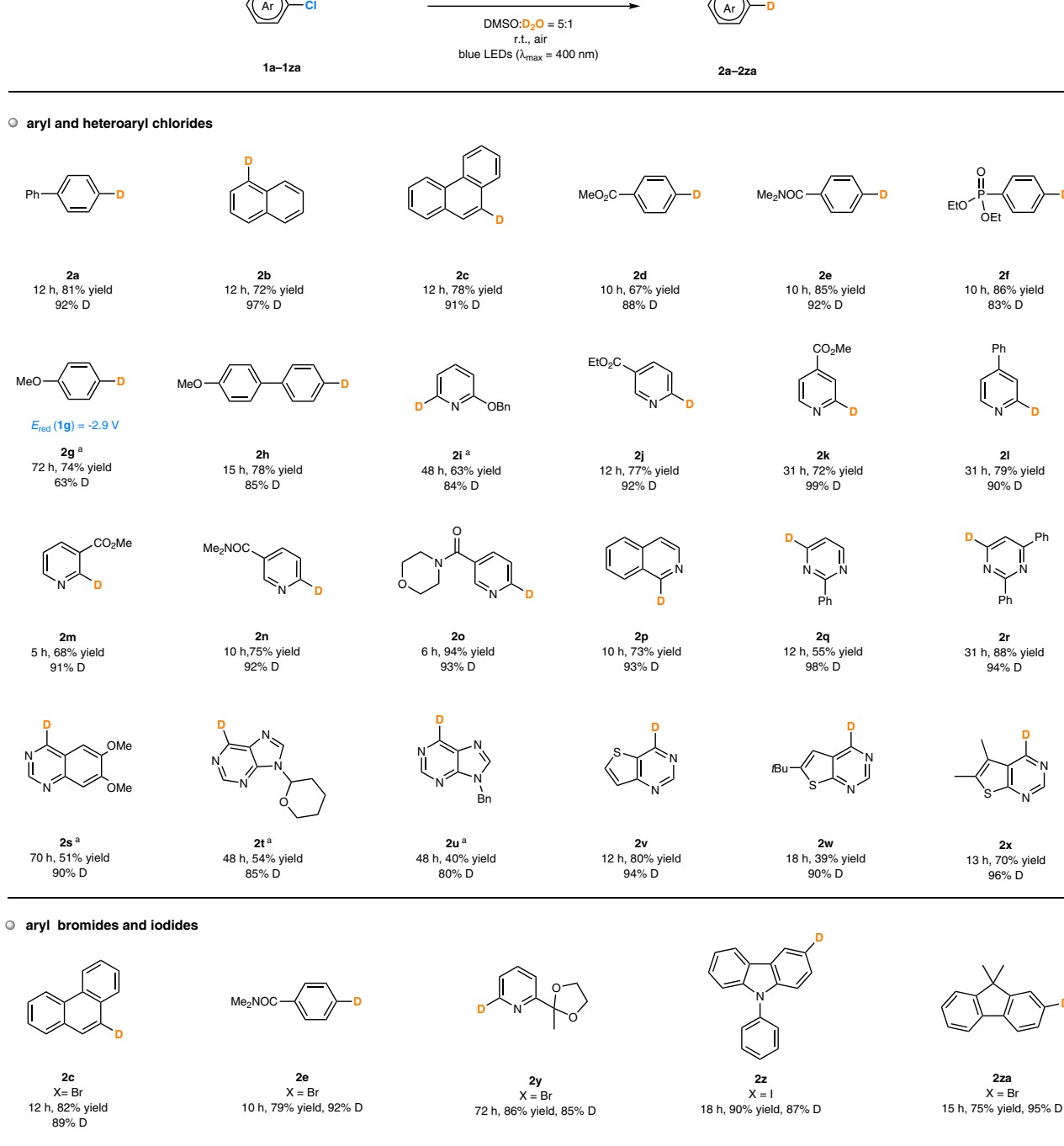

**Fig. 4 Scope of photocatalytic deuterodehalogenation of aryl chlorides and other halides.** [a]**PC4** (20 mol%), ($n$PrS)$_2$ (30 mol%), in argon. PC, photocatalyst; r.t., room temperature; eq, equivalent; DMSO, dimethyl sulfoxide.

isoquinoline (product **2p**), pyrimidines (products **2q**, **2r**), quinazoline (product **2s**), purines (products **2t**, **2u**), thieno[3,2-$d$] pyrimidine (product **2v**) and thieno[2,3-$d$]pyrimidines (products **2w**, **2x**) were all obtained in reasonable yields (39–94%) and with excellent deuterium incorporation (80–99%). These results also revealed the excellent functionality compatibility of the reaction, which could be useful in its further application to structurally more complex pharmaceutical molecules and drugs[35]. Moreover, this method was applicable to other aryl halides. For example, the D-labeled products (**2c**, **2e**, **2y**–**2za**) could be obtained by the reaction of the corresponding aryl bromides or iodides under the standard conditions, providing similar deuterium incorporation while increasing the reaction rates.

**Deuterodehalogenation of alkyl chlorides.** Unactivated alkyl chlorides are particularly resistant to deuterodehalogenation. The traditional transition-metal catalytic approaches are not applicable due to the possibility of side reactions such as β-elimination. The highly negative reduction potentials of C(sp$^3$)-Cl bonds and the reactive intermediates eliminate them from the scope of radical-mediated methods[65]. For these reasons,

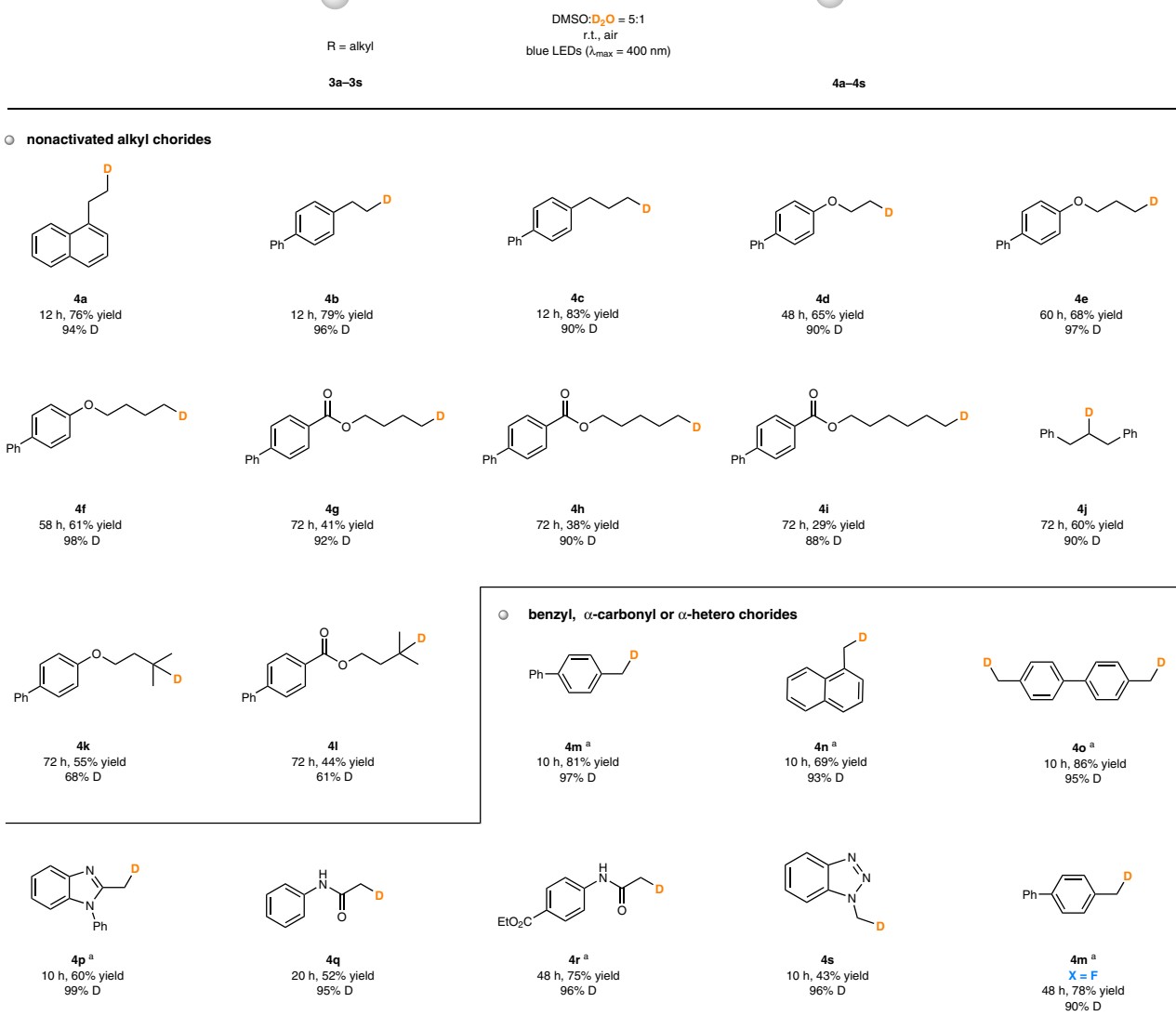

**Fig. 5 Scope of photocatalytic deuterodehalogenation of alkyl chlorides.** [a]**PC4** (10 mol%). PC, photocatalyst; r.t., room temperature; eq, equivalent; DMSO, dimethyl sulfoxide.

deuterodehalogenation of unactivated alkyl chlorides has not been reported to date.

To our delight, the photochemical deuterodehalogenation reaction of **3a**, a terminal alkyl chloride, gave the desired deuterated product (**4a**) in 76% yield and with 94% deuterium incorporation within 12 h in the presence of **PC4** ($E^*_{ox} = -2.42$ to $-2.60$ V vs SCE) as the photocatalyst (see more details for reinvestigation of reaction conditions in Supplementary Table 2). A number of unactivated alkyl chlorides were examined under the developed reaction conditions (Fig. 5). Primary alkyl chlorides bearing a remote aromatic ring (products **4a–4c**), an ether bond (products **4d–4f**) or an ester linkage (products **4g–4i**) all provided the deuterated products in reasonable yields (29–83%) and with high deuterium incorporation (88–98%). The secondary alkyl chloride also reacted, delivering product **4j** in 60% yield and with 90% D-inc. In comparison, the tertiary alkyl chlorides were less effective substrates, whose reactions proceeded much more slowly and provided **4k**, **4l** with 68 and 61% D-inc., respectively. Relatively active C(sp³)-Cl precursors containing a benzyl group (products **4m–4p**), an α-carbonyl moiety (products **4q**, **4r**) and

α-hetero substitutent (products **4s**) were found to be excellent substrates in photocatalytic deuterodehalogenation. These reactions proceeded more rapidly (typically at a lower catalyst loading) and resulted in higher deuterium incorporation (93–99%). The neighboring effects even enabled the reductive cleavage of C-F followed by deuteration, affording product **4m** from a fluorine-containing starting material.

**Site-selective deuterodehalogenation of polychlorinated compounds.** Selective funtionalization of carbon-halogen bonds with little or no difference in polarity and other features in poly-halogenated compounds is a highly desirable attribute in multi-step organic synthesis but remains highly challenging[66–69]. It was found that this photocatalytic system could selectively install one deuterium atom in dichlorinated or polychlorinated compounds through choice of a photocatalyst with appropriate reducing ability, that is important for rapid construction of deuterated molecule arrays through diverse functionalization of the residual C-Cl bonds (Fig. 6). For example, under the standard conditions and in the presence of **PC1** or **PC2** as the photocatalyst, the

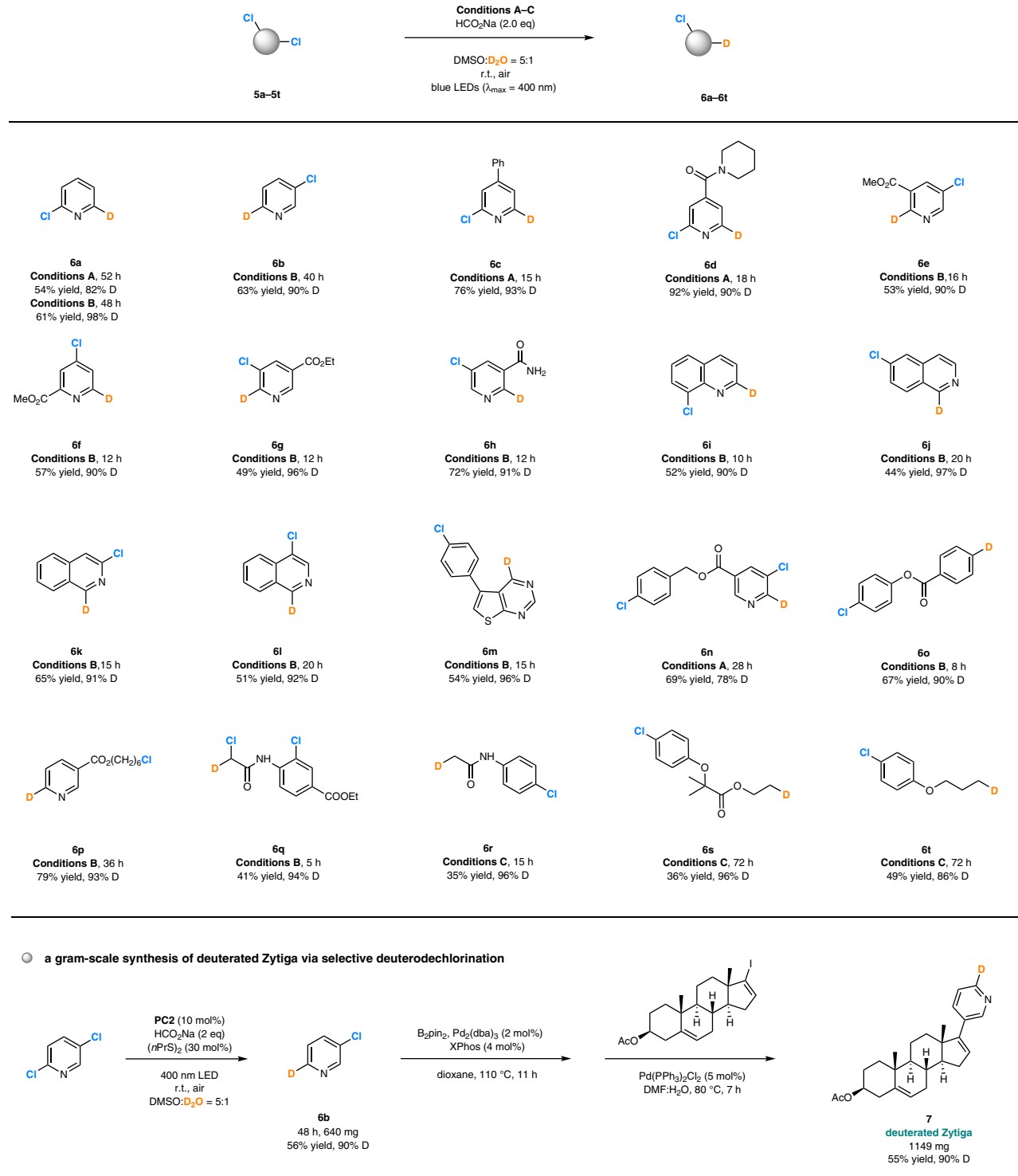

**Fig. 6 Site-selective deuterodehalogenation of polychlorides. Conditions A:** PC1 (10 mol%), (MeS)₂ (30 mol%). **Conditions B:** PC2 (10 mol%), (nPrS)₂ (30 mol%). **Conditions C:** PC4 (20 mol%), (nPrS)₂ (30 mol%). PC, photocatalyst; r.t., room temperature; eq, equivalent; DMSO, dimethyl sulfoxide; DMF, N,N-dimethylformamide; B₂pin₂, Bis(pinacolato)diboron.

reactions of symmetric dichlorinated pyridine derivatives (**5a–5d**) only afforded monodeuterated compounds (**6a–6d**) in 61–92% yield and with 90–98% D-inc. Prolonged reaction time did not lead to the formation of dideuterated products. Excellent site-selectivity could be observed for a number of unsymmetric (hetero)aryl chlorides, all of which provided monodeuterated products (**6e–6o**) with high deuterium incorporation (78–97%). Typically, a C-Cl bond at α-positions next to the nitrogen atom in

pyridines, diazines or quinolines was more readily to be reacted, which is a hot spot for aldehyde oxidase (AOX) metabolism[70]. Assemble of deuterium atoms at these positions in drugs would be useful for pharmacokinetic investigations. Moreover, selective deuterodehalogenation of compounds bearing both C(sp²)-Cl and C(sp³)-Cl bonds could be accessed as well. For example, photocatalyst **PC2** enabled attachment of one deuterium atom to the pyridyl moiety of **5p** and delivered product **6p** with 93%

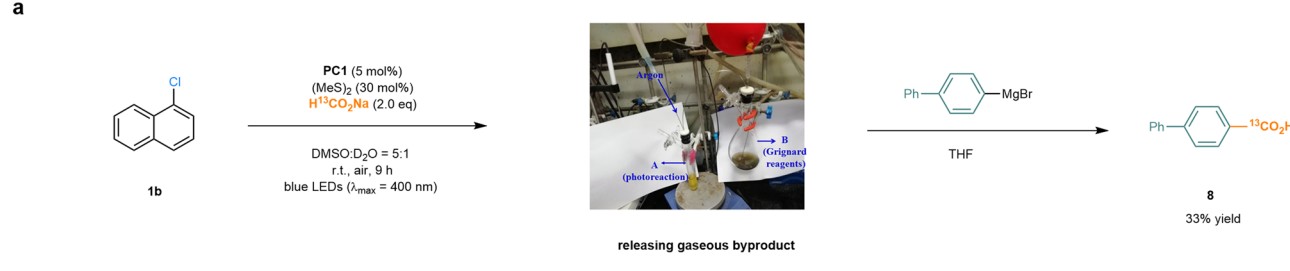

**Fig. 7 Mechanistic studies. a** $CO_2$ trapping experiments. **b** Stern–Volmer quenching experiments. **c** Thiyl radical scrambling experiments. **d** Cyclic voltammetry of the dichloride and the possible mono reduction products. PC, photocatalyst; r.t., room temperature; eq, equivalent; THF, tetrahydrofuran; DMSO, dimethyl sulfoxide; DFT, density functional theory; BDE, bond dissociation energy; SCE, saturated calomel electrode.

D-inc., while photocatalyst **PC4** offered an opportunity to introduce deuterium atoms on the alkyl chains instead of on the electron-rich phenyl rings in **5r**−**5t** (86-96% D-inc.). Such a transformation has not been realized by other reported methods, and will provide a useful access to selectively incorporate deuterium atoms in complex halogen-containing biomolecules as well as drugs. For example, Zytiga is a specific medicine for the treatment of patients with metastatic castration-resistant prostate cancer or high-risk

castration-sensitive prostate cancer[71], and was the 46th drug in retail sales in 2019[72]. A gram-scale synthesis of D-labeled Zytiga **7** (1149 mg, 90% D-inc.) could be readily developed based on the selective deuterodehalogenation of 2,5-dichoropyridine (product **6b**) followed by Pd-catalyzed coupling reactions.

**Mechanistic studies**. Several control experiments were conducted to gain some insight into the mechanism of this reaction (Fig. 7).

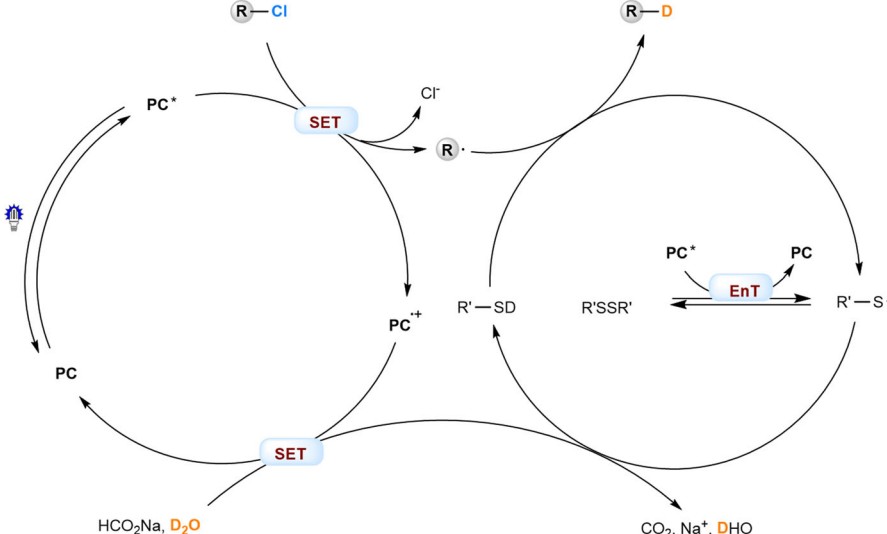

**Fig. 8 Proposed reaction mechanism.** The photocatalysts are introduced as strongly reducing photoredox catalysts to initiate cleavage of C-Cl bonds, at the same time as energy transfer catalysts to induce homolysis of the disulfide cocatalysts. PC, photocatalyst; SET, single electron transfer; EnT, energy transfer.

The photochemical reaction **1b**→**2b** under the standard conditions with the $^{13}C$-labeled sodium formate (H$^{13}CO_2Na$) led to the formation of $^{13}CO_2$, which was trapped by a Grignard reagent to afford the $^{13}C$-containing *p*-phenylbenzoic acid (**8**) in 33% yield (Fig. 7a). This observation confirmed that sodium formate serves as an essential electron donor (a reductant) in the catalytic cycles with release of $CO_2$ as a byproduct. Luminescence quenching experiments revealed that the aryl chloride (**1b**) was capable of obviously quenching the excited state of **PC1** (Fig. 7b, left). The disulfide cocatalyst (MeS)$_2$ somewhat quenched its excited state as well while sodium formate failed to do so, revealing the possible interactions between the photocatalyst and other reaction components. The similar results were observed in a system of photocatalyst **PC4**, alkyl chloride **3b**, 1,2-dipropyldisulfane and sodium formate (Fig. 7b, right).

A radical scrambling experiment was performed to probe the existence of thiyl radical intermediates in the photochemical reaction. Under the standard conditions and in the presence of the photocatalyst **PC4**, irradiation of dipropyl disulfide and dicyclohexyl disulfide with blue light led to the observation by HRMS of cyclohexyl propyl disulfane (Fig. 7c). The formation of cyclohexyl propyl disulfane probably involved thiyl radicals as intermediates and saw their radical recombination or radical addition into another disulfide molecule. Density functional theory calculations further established that an energy transfer between alkyl disulfides (BDE (S–S) = 65.0 kcal mol$^{-1}$) and the photocatalyst ($E_T$ (**PC4**) = 65.5 kcal mol$^{-1}$) was energetically favorable in the reaction (see more details in Supplementary Method 4.8)[73].

Electrochemical analysis of one dichlorinated starting material (**5l**), H-analogs of the possible regioisomeric products (**6l′**, **6l″**) was performed (Fig. 7d). Two reversible reduction/oxidation signals were observed at $E_{red} = -2.3$ V and $-2.0$ V for **5l**. The reversible reduction peaks of **6l′** and **6l″** were found at $-2.4$ V and $-2.1$ V, respectively. These outcomes suggested that C-Cl bond at α-position of the isoquinoline moiety was more active than that at the other position, and **6l′** was more resistant to further reductive deuterodehalogenation. However, the electronic properties and bond dissociation energies (BDE (C-Cl) = 88.7 vs 84.3 kcal mol$^{-1}$) are close to each other, demonstrating that this photocatalytic system has precise recognition ability toward these

similar C-Cl bonds (see more details in Supplementary Method 4.9).

**Mechanistic proposal**. On basis of the initial experiments and these mechanistic studies, a plausible mechanism for the photocatalytic deuterodehalogenation of organochlorides is shown in Fig. 8. The photoexcited catalyst (**PC***, $E^*_{ox} = -1.61$ to $-2.71$ V vs SCE) undergoes single electron transfer with the organic chloride ($E_{red} = -1.8$ to $-3.0$ V) to give rise to **PC**$^{\cdot+}$ and a carbon radical (R•). Meanwhile, **PC*** serve as an energy transfer catalyst ($E_T$ (**PC4**) = 65.5 kcal mol$^{-1}$) to induce homolysis of the disulfides (BDE (S-S) = 65.0 kcal mol$^{-1}$) affording a thiyl radical (R'S•). In combination with sodium formate as the electron donor and hydrogen donor, R'S• performs effective deuterium atom transfer, and close both the photocatalytic and organocatalytic cycles.

**Synthetic application to D-labeled bioactive molecules and drugs**. Since the developed photocatalytic system exhibits excellent functional group compatibility and broad substrate scope, we questioned whether it could be used for deuterium-labeling of more complex bioactive molecules and drug derivatives (Fig. 9). For example, Etofibrate is a compound produced by the combination of clofibrate ester linked to niacin, and is used to treat hyperlipemia[74]. Photocatalytic deuterodehalogenation of its chlorinated derivatives with **PC1** as the photocatalyst under the standard conditions led to selective incorporation of deuterium atoms into Etofibrate at specific positions of the pyridyl moiety without affecting the chlorine atoms on the electronically rich phenyl ring, and delivered deuterated products (**9**, **10**) with 91 and 90% D-inc., respectively. Using a similar strategy, deuterated Nikethamide (**11**), Pyriproxifen (**12**), and Nicoboxil (**13**) were obtained in good yields (64−86%) and with high deuterium incorporation (86−99%). Boscalid from BASF is a nicotinamide germicide, which has a broad spectrum of bactericidal activity and has a preventative effect, being active against many types of fungal diseases[75]. Under the standard conditions, the C(sp$^2$)-Cl bond located on the pyridyl moiety of Boscalid was selectively converted into a C-D bond, affording a product (**14**) in 89% yield and with 88% deuterium incorporation. Deuterodehalogenation

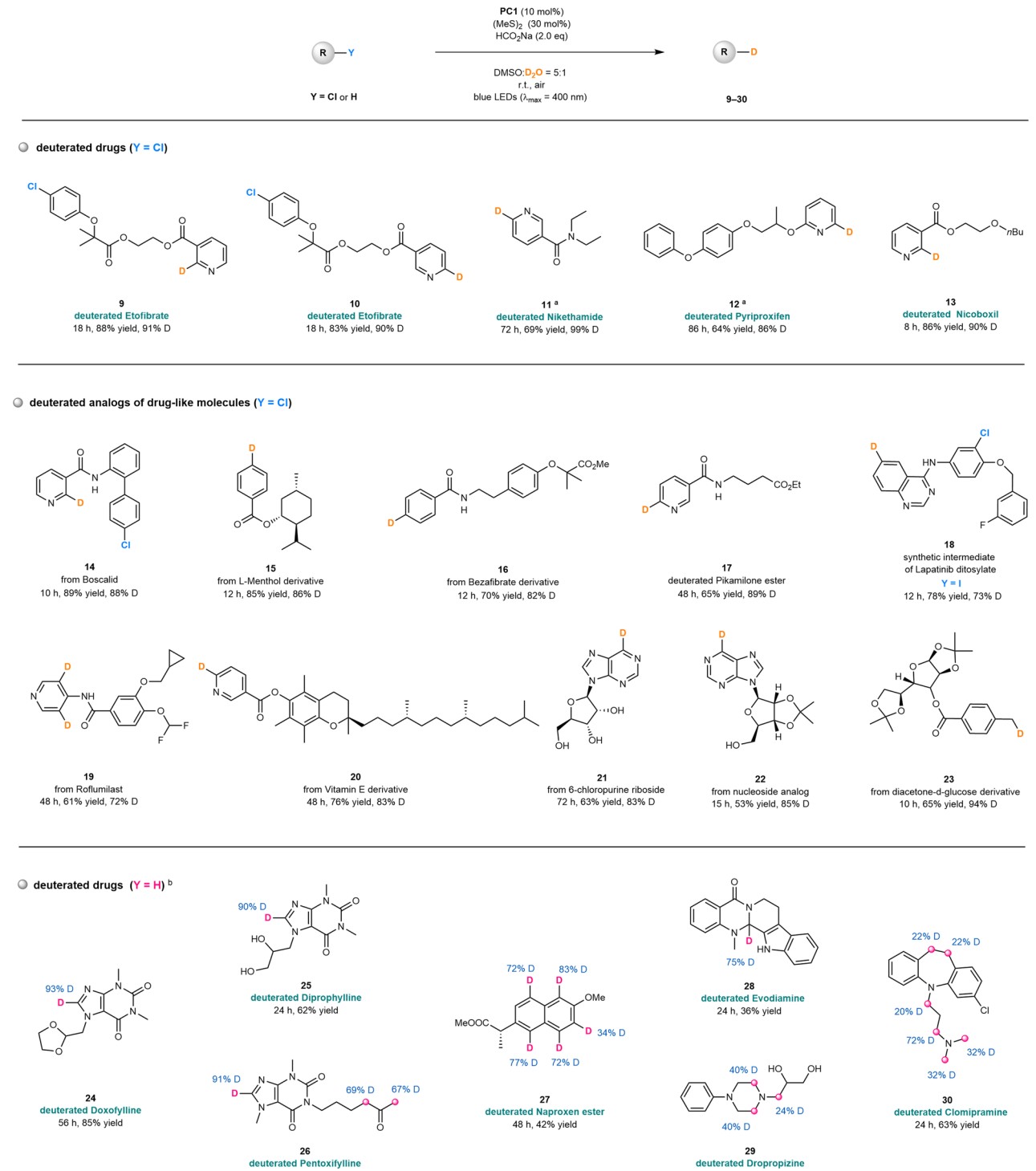

**Fig. 9 Synthetic application to D-labeled bioactive molecules and drugs.** [a]**PC4** (20 mol%), ($n$PrS)$_2$ (30 mol%), in argon. [b]**PC2** (10 mol%), ($n$PrS)$_2$ (60 mol%), HCO$_2$Na (4.0 eq). PC, photocatalyst; eq, equivalent; r.t., room temperature; DMSO, dimethyl sulfoxide.

of an L-menthol derivative (product **15**), Bezafibrate derivative (product **16**), Pikamilone ester (product **17**), a synthetic intermediate of Lapatinib ditosylate (Y = I, product **18**), Roflumilast (product **19**), Vitamin E derivative (product **20**), 6-chloropurine riboside (product **21**), and nucleoside analog (product **22**) all proceeded smoothly, leading to the formation of deuterated products in 53−89% yield and with 72−88% D-inc. The diacetone-d-glucose derivative, an aliphatic chlorinated

biomolecule, was also found to be well tolerated in the reaction and provided product **23** in 65% yield and 94% D-inc.

H/D exchange of C(sp$^3$)-H or C(sp$^2$)-H bonds is a appealing access to deuterated molecules, but its application in D-labeling of pharmaceutical compounds remains a formidably difficult problem[25]. Interestingly, this photocatalytic system could be used for direct H/D exchange of some drug and drug-like molecules. Under modified conditions in the presence of **PC2** as

the photocatalyst, the C(sp²)-H bond located on the imidazole moiety of Doxofyline, an injectable drug for treatment of asthma, was selectively deuterated, delivering product **24** in 85% yield and 93% D-inc. within 56 h. The reactions of its derivatives, Diprophylline and Pentoxifylline, gave similar results and afforded product **25** and **26** with good deuterium incorporation, respectively. Such H/D exchange phenomena was observed in the photochemical reactions of Naproxen ester (product **27**), Evodiamine (product **28**), Dropropizine (product **29**), and Chomipramine (product **30**), in which one or more C(sp²)-H/C(sp³)-H bonds were deuterated. These outcomes further demonstrated synthetic utility of this method in D-labeling of structurally complex biomolecules.

## Discussion

We have designed an organophotocatalytic system by merging arylamine-based photocatalysis and disulfide catalysis. The conjugate aryl amines, such as **PC1** and **PC2** which have not been used as photocatalysts in previous studies, are introduced as strongly reducing photoredox catalysts to initiate cleavage of C-Cl bonds, at the same time as energy transfer catalysts to induce homolysis of the disulfide cocatalysts. Common alkyl disulfides are involved to enable a high level of deuterium incorporation. Sodium formate is employed as a mild electron and hydrogen sacrificial agent to promote the radical transformations and to close the catalytic cycles. This strategy allows us establish an efficient channel for deuterium transfer, and suppressing side reactions such as self-coupling and elimination. Under very mild reaction conditions (visible light, D₂O/DMSO, room temperature, in air, benign byproducts such as $CO_2$), a wide range of aryl chlorides, alkyl chlorides, and other halides including many bioactive molecules (>90 examples) can be converted to deuterated products in good yields and with up to 99% D-incorporation. Moreover, precise site-selectivity toward C-Cl bonds of polychlorinated compounds that have little or no difference in polarity or other features has been observed. The successful production to deuterated drugs and other bioactive molecules as well as application to H/D exchange of C(sp²)-H or C(sp³)-H bonds with D₂O demonstrate considerable potential of this method in pharmaceutical chemistry.

## Methods

**General procedure for deuterodehalogenation of aryl chlorides**. A dried 5 mL glass vial was charged with aryl chlorides (0.20 mmol), photocatalyst **PC1** (8.7 mg, 0.020 mmol), (MeS)₂ (6 μL, 0.060 mmol), HCO₂Na (27.2 mg, 0.40 mmol), D₂O (200 μL) and DMSO (1.0 mL) under air and then performed in a sealed vessel. The glass vial was positioned approximately 3 cm away from a 50 W blue LEDs lamp ($\lambda_{max}$ = 400 nm). After being stirred at room temperature (~30 °C under irradiation) for the indicated time, the reaction mixture was purified by flash chromatography on silica gel to afford product.

**General procedure for deuterodehalogenation of alkyl chlorides**. A dried 5 mL glass vial was charged with alkyl chlorides (0.20 mmol), photocatalyst **PC4** (11.3 mg, 0.040 mmol), (nPrS)₂ (10 μL, 0.060 mmol), HCO₂Na (27.2 mg, 0.40 mmol), D₂O (200 μL) and DMSO (1.0 mL) under air and then performed in a sealed vessel. The glass vial was positioned ~3 cm away from a 50 W blue LEDs lamp ($\lambda_{max}$ = 400 nm). After being stirred at room temperature (~30 °C under irradiation) for the indicated time, the reaction mixture was purified by flash chromatography on silica gel to afford product.

## Data availability

The authors declare that the data supporting the findings of this study are available within this article and its Supplementary Information file, or from the corresponding author upon reasonable request. The experimental procedures and characterization of all new compounds are provided in Supplementary Information.

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

## Acknowledgements

We gratefully acknowledge funding from the National Natural Science Foundation of China (grant no. 22071209, 22071206), the Natural Science Foundation of Fujian Province of China (grant no. 2017J06006), and the Fundamental Research Funds for the Central Universities (grant no. 20720190048).

## Author contributions

Y.L. and L.G. designed and conceived the project. Y.L. and Z.Y. conducted all the synthetic and catalytic reactions. Y.L., Z.Y., Y.-M.L., and L.G. analyzed and interpreted the experimental data. Z.Y. designed and performed the theoretical calculations. Y.L. and Y.Z. performed the HRMS experiments and analyzed the data. L.G. prepared the manuscript. Y.L. and Z.Y. prepared the Supplementary Information.

## Competing interests

The authors declare no competing interests.
