## [Peer Review File · Nature Communications]

REVIEWER COMMENTS

Reviewer #1 (Remarks to the Author):

Significance.

This manuscript by Prof. Lei and his co-workers reports an effective method to synthesize deuterated (hetero)arenes and alkanes. Notably, this method avoids using precious metal reagent/catalyst, complex ligand, special/expensive deuterium sources and harsh conditions. Although several radical mediated dehalogenative deuteration of organohalides were previously reported (ref. 40, 43), his method allows the use of abundant chlorides, which is a big-step advance of the state-of-the-art methodologies. It is even remarkable that this method can bias the two Cl atoms on the aryl core with only slightly different environments, leading to the precise incorporation of the D labeling. At the last part, the authors also showcased the capability of their method to direct H/D exchange of sp² C–H and activated sp³ C–H. The deuterated compounds can be effectively generated by this method are of great interest of various research areas, such as organic synthesis, mechanistic study, quantitative analysis and drug development as the authors have mentioned and demonstrated in the manuscript. Therefore, this manuscript is of great significance in both the underlying chemistry and the potential applications. Based on those points, I value this manuscript as an excellent contribution and I support its publication in Chem.

Minors:

1. The Loh's photocatalytic dehalogenative deuteration chemistry should be highlighted in the introduction section to show an overall techniques in the area. Loh's work using porous CdSe nanosheets as the catalyst, is fundamentally different with the chemistry described here, which does not diminish the value of this contribution.
2. Line 8, page 1 "tend" should be "tends".
3. The references cover the key papers reported in or related to the field. There is one key paper about dehalogenative deuteration of chloride missing: Kuriyama, M.; Hamaguchi, N.; Yano, G.; Tsukuda, K.; Sato, K.; Onomura, O. *J. Org. Chem.* 2016, 81, 8934
4. The SI was well organized and presented in a professional way, and most of the spectra are in high quality. There are several typos to be addressed, 1) page 1, 9, "spectrum" should be "spectra"; 2) S105, for the product 18, Y = I, not X = I. The C NMR of this product is mis-assigned (rotamer?).

Reviewer #2 (Remarks to the Author):

Authors describe their design of a new organophotocatalytic system to enable a selective deuterodechlorination of aryl or alkyl chlorides. Since the reaction smoothly proceeded to afford useful deuterated products in a high site-selectivity under mild conditions, many readers will take strong interests in their strategy. Also, the mechanistic studies seem to be plausible. However, values of D-incorporation in most deuterated products are insufficient for use in pharmaceuticals.

Reviewer #3 (Remarks to the Author):

Gong and co-workers reported a photoinduced organo-photosensitizer and disulfide co-catalyzed deuterodehalogenation of aryl chlorides with good yields and excellent D-incorporation at room temperature in air. This method shown good tolerance of functional groups, as well as good site-selectivity for polychlorinated compounds. It is interesting that this method can be extend to alkyl chlorides. The reviewer recommended to published in this journal. There are some questions need to answer:

- (1) The proposal mechanism need to further investigate. The R radical possibly further reduce to form R anion (rather than abstract H radical from thiol), then abstract proton form D₂O, especially for the alkyl chloride substrates.
- (2) What is the by-products for the reaction of 3g? It is possible to detect ketone by-product which prove the anion intramolecular attack the ester.
- (3) How about directly D/H exchange of the Zytiga 7 to obtain deuterated Zytiga in standard condition?

Point-by-Point Response to the Reviewers' Comments

Responses to the comments of reviewer 1.

(1) This manuscript by Prof. Lei and his co-workers reports an effective method to synthesize deuterated (hetero)arenes and alkanes. Notably, this method avoids using precious metal reagent/catalyst, complex ligand, special/expensive deuterium sources and harsh conditions. Although several radical mediated dehalogenative deuteration of organohalides were previously reported (ref. 40, 43), his method allows the use of abundant chlorides, which is a big-step advance of the state-of-the-art methodologies. It is even remarkable that this method can bias the two Cl atoms on the aryl core with only slightly different environments, leading to the precise incorporation of the D labeling. At the last part, the authors also showcased the capability of their method to direct H/D exchange of sp^2 C–H and activated sp^3 C–H. The deuterated compounds can be effectively generated by this method are of great interest of various research areas, such as organic synthesis, mechanistic study, quantitative analysis and drug development as the authors have mentioned and demonstrated in the manuscript. Therefore, this manuscript is of great significance in both the underlying chemistry and the potential applications.....

Our response:

We sincerely thank the positive evaluation on our manuscript and encouragement from reviewer 1. The comments and suggestions inspire us to further improve the manuscript.

(2) The Loh's photocatalytic dehalogenative deuteration chemistry should be highlighted in the introduction section to show an overall techniques in the area. Loh's work using porous CdSe nanosheets as the catalyst, is fundamentally different with the chemistry described here, which does not diminish the value of this contribution.

Our response:

Thanks for the highly valuable suggestion. In the revised manuscript, we have added the statement "Loh *et al.* developed the photocatalytic deuterodehalogenation of aryl halides using porous CdSe nanosheets as the catalyst and D_2O as the deuteration

reagent.” to the introduction section regarding this excellent work (page 2).

(3) Line 8, page 1 “tend” should be “tends”.

Our response:

It has been corrected in the revised manuscript (page 1). Moreover, we have carefully checked the manuscript to avoid this type of mistakes.

(4) The references cover the key papers reported in or related to the field. There is one key paper about dehalogenative deuteration of chloride missing: Kuriyama, M.; Hamaguchi, N.; Yano, G.; Tsukuda, K.; Sato, K.; Onomura, O. *J. Org. Chem.* 2016, 81, 8934.

Our response:

Thanks for the suggestion, we have added this important reference as ref 46 (page 26).

(5) The SI was well organized and presented in a professional way, and most of the spectra are in high quality. There are several typos to be addressed, 1) page 1, 9, “spectrum” should be “spectra”; 2) S105, for the product **18**, Y = I, not X = I. The C NMR of this product is mis-assigned (rotamer?).

Our response:

We apologize for our carelessness. It has been corrected in the revised SI (page S107). Moreover, we have carefully checked the SI to avoid this type of mistakes.

Responses to the comments of reviewer 2.

(1) Authors describe their design of a new organophotocatalytic system to enable a selective deuterodechlorination of aryl or alkyl chlorides. Since the reaction smoothly proceeded to afford useful deuterated products in a high site-selectivity under mild conditions, many readers will take strong interests in their strategy. Also, the mechanistic studies seem to be plausible. However, values of D-incorporation in most

deuterated products are insufficient for use in pharmaceuticals.

Our response:

We sincerely thank the positive evaluation on our manuscript and encouragement from reviewer 2. The comments and suggestions inspire us to further improve the manuscript.

Responses to the comments of reviewer 3.

(1) Gong and co-workers reported a photoinduced organo-photosensitizer and disulfide co-catalyzed deuterodehalogenation of aryl chlorides with good yields and excellent D-incorporation at room temperature in air. This method shown good tolerance of functional groups, as well as good site-selectivity for polychlorinated compounds. It is interesting that this method can be extend to alkyl chlorides. The reviewer recommended to published in this journal.

Our response:

We sincerely thank the highly positive evaluation on our manuscript and encouragement from reviewer 3. The comments and suggestions inspire us to further improve the manuscript.

(2) The proposal mechanism need to further investigate. The R radical possibly further reduce to form R anion (rather than abstract H radical from thiol), then abstract proton form D₂O, especially for the alkyl chloride substrates.

Our response:

Thanks for the highly valuable suggestion. It encourages us to look into the reaction system in more depth. Indeed, pathways involving R anions are possible in many redox process. However, it is most likely not the case in our photochemical system. In this deuterodehalogenation reaction, CO₂ was detected as a byproduct while carboxylic acids could not, suggesting that anion mechanism might not be the main pathway. Moreover, small amount of side-products such as cross-coupling products (R-SR') could be isolated (see below, reaction of **3g**), which also support the hypothesis of a radical pathway.

(3) What is the by-products for the reaction of **3g**? It is possible to detect ketone by-product which prove the anion intramolecular attack the ester.

Our response:

We isolated the byproducts of the reaction of **3g**, which were identified as a cross-coupling product **4g'** (R-SR', 6% yield) and **4g''** (~ 20% yield). The structure of **4g''** was unclear, but it was apparently not the by-product formed via anion intramolecular attack of the ester (see ¹H NMR and ¹³C NMR of compound **4g''** below). These outcomes further establish that anion mechanism might not be the main pathway. We add the results in the revised supplementary information (page S48 and S203).

¹H NMR
CDCl₃

(4) How about directly D/H exchange of the Zytiga **7** to obtain deuterated Zytiga in standard condition?

Our response:

The direct D/H exchange of Zytiga **7** was examined under the standard conditions, however, no deuterated Zytiga was obtained. We assumed that D/H exchange could only occurred on the relatively activated C-H bonds. We add the result in the revised supplementary information (page S120).

REVIEWERS' COMMENTS

Reviewer #3 (Remarks to the Author):

The authors have answered all of the reviewer's mentioned questions. Thus, the reviewer recommend to publish in this journal.